# Community responses to a novel house design: A qualitative study of "Star Homes" in Mtwara, southeastern Tanzania

Salum Mshamu[1,2☉], Judith Meta[3☉], Bipin Adhikari[2,4☉]*, Salma Halifa[1], Arnold Mmbando[5,6], Hannah Sloan Wood[7], Otis Sloan Wood[7], Thomas Chevalier Bøjstrup[7], Nicholas P. J. Day[2,4], Steven W. Lindsay[6], Jakob Knudsen[7], Jacqueline Deen[8], Lorenz von Seidlein[2,4], Christopher Pell[9,10,11]

1 CSK Research Solutions, Mtwara, Tanzania, 2 Centre for Tropical Medicine and Global Health, Nuffield Department of Medicine, University of Oxford, Oxford, United Kingdom, 3 Private Consultant, Social Scientist, Mtwara, Tanzania, 4 Mahidol Oxford Tropical Medicine Research Unit, Faculty of Tropical Medicine, Mahidol University, Bangkok, Thailand, 5 Ifakara Health Institute, Ifakara, Tanzania, 6 Department of Biosciences, Durham University, Durham, United Kingdom, 7 Royal Danish Academy – Architecture, Design, Conservation, Copenhagen, Denmark, 8 University of Philippines, Manila, Philippines, 9 Amsterdam University Medical Centres, University of Amsterdam, Department of Global Health, Amsterdam, The Netherlands, 10 Amsterdam Institute for Global Health and Development, Amsterdam, The Netherlands, 11 Amsterdam Public Health Research Institute, Amsterdam, The Netherlands

☉ These authors contributed equally to this work.
* Bipin@tropmedres.ac

**Data Availability Statement:** Because of the nature of the qualitative data in this study, even if the data are anonymized, potential respondents are identifiable. Both MORU and local ethics committee

## Abstract

### Introduction

To evaluate the impact of a novel design "Star Home" on the incidence of malaria, respiratory tract infections and diarrheal diseases among children, randomly selected households in Mtwara, Tanzania were offered a free, new Star Home. Drawing on longitudinal qualitative research that accompanied the Star Homes study, this article describes the experiences of residents and the wider community of living with these buildings.

### Methods

A total of four rounds of face-to-face interviews were undertaken with residents of Star Homes (n = 37), control (wattle/daub) homes (n = 21), neighboring households n = 6), community members (n = 17) and community leaders (n = 6). The use of Star Homes was also observed over these four time periods between 2021 and 2023. Interviews were conducted in Swahili, transcribed, and translated into English for thematic analysis.

### Results

Star Homes residents appreciated several aspects of the Star Homes, including overall comfort, access to water and electricity, and clean toilets. There were concerns about some design elements, such as poorly closing doors, stoves perceived as inefficient, and the façade, which was susceptible to rainwater ingress. The houses were not always used as intended by their developers, for example, residents were sleeping downstairs instead of

restricts the sharing of data that can potentially identify the respondents. The data is available upon request to the Mahidol Oxford Tropical Medicine Research Unit Data Access Committee (datasharing@tropmedres.ac) complying with the data access policy on case-by-case analysis (https://www.tropmedres.ac/units/moru-bangkok/bioethics-engagement/data-sharing/moru-tropical-network-policy-on-sharing-data-and-other-outputs).

**Funding:** This study was funded by Hanako foundation, Singapore. The funders had no role in study design, data collection and analysis, decision to publish, or preparation of the manuscript.

**Competing interests:** The authors have declared that no competing interests exist.

**Abbreviations:** ARIs, acute respiratory tract infections; COVID-19, Corona Virus Infectious Diseases-2019; ITN, Insecticide Treated bed Nets; SDG, Sustainable Development Goals; SSA, Sub Saharan Africa.

upstairs because of cold floors or difficulties using the stairs. Star Homes residents described how the structures triggered praise but also envy from other community members.

## Conclusions

The findings highlight the need for close attention to the use of novel design houses and careful sensitization around the potential benefits of dwellings to ensure that the intended health impacts of interventions are achieved.

## Introduction

Malaria, acute respiratory tract infections (ARIs), and diarrheal diseases are responsible for more than a third of all childhood deaths in sub-Saharan Africa [1–3]. Nearly five children under the age of five years old die every minute due to these preventable causes of deaths [4]. In sub-Saharan Africa, around 80% of all malaria deaths occur among under-five children: 400,000 deaths annually [5]. Together with Nigeria, Democratic Republic of the Congo, and Niger, The United Republic of Tanzania was among the countries that accounted for over half of global malaria deaths in 2022 [6].

Although many of these infections can be treated in healthcare facilities, in places where there are strained healthcare systems and limited access to care, community-based interventions are considered key to reducing morbidity and mortality [7, 8]. Environmental factors are major contributors to the transmission of infectious diseases—unsafe water, sanitation and hygiene, and indoor air pollution—remain major risks for health and well-being globally [9–11]. Indeed, many of the greatest advances in human health have focused on addressing the environmental causes of infectious diseases, particularly through cleaner water and improved sanitation [12].

The large-scale distribution of insecticide treated bed nets (ITNs) around the turn of the 21st century had an unprecedented impact on malaria transmission in sub-Saharan Africa. More recently, however, this trend has reversed [13, 14]. Part of this is likely a result of the poor coverage of ITNs, deterioration of the bio-efficiency and quality of ITNs, and reticence to sleep under the ITNs in the high indoor temperatures of many African homes [15–17]. The periodic distribution of ITNs also raises environmental concerns around production, distribution, and disposal including its contribution to plastic pollution [18]. A more sustainable approach to reducing contact with malaria vectors is therefore needed.

Housing interventions are a promising long-term strategy to reduce contact with malaria vectors and to prevent other infectious diseases, in a comfortable living environment that ultimately can improve wellbeing [19, 20]. Observational studies found that, living in improved housing offered a 47% lower risk of malaria infection, with 45–65% fewer malaria cases compared to traditional homes [20, 21]. Although most housing projects make modifications of existing structures, researchers have begun to explore the impact of entire 'novel design homes' on disease transmission [22]. Given the key role that housing plays in our everyday lives and its embedding in specific social, cultural, and environmental contexts, the practicality and acceptability of whole-house interventions are essential pre-conditions for scaling up and roll-out of this new intervention [10]. Human dwellings are coded by cultural traditions, historical developments, the local ecology, environment, personal preferences, and wealth [23].

Their design and construction also have important ramifications on how people perform their daily practices [24].

In Magoda, northeastern Tanzania, a pilot project demonstrated the feasibility and acceptability of novel house designs [25], which have been in use without interruption or significant reconfiguration. More research is needed to generate evidence on the health impact and acceptability of whole-house interventions. The Star Homes Project, a multi-disciplinary collaboration, aims to address this gap by developing and evaluating a novel house design in Mtwara Region, southeastern Tanzania [20, 26].

The Star Home design (Fig 1) was developed to protect against malaria, ARIs, and diarrhoeal diseases [22, 27–29]. The design process utilized an iterative approach of prototyping, evaluation, and improvement. The project team created a prototyping site in the Mtwara town, to test different design iterations, construction methods, building components, and to gather feedback prior to construction of the 110 houses. The principal evaluation is a randomized controlled trial comparing the incidence of the three diseases in children under 13 years living in Star Homes compared with traditional wattle-daub houses. Social science studies support the main study by assessing the acceptability and feasibility of the study [22].

A total of 110 Star Homes were built across 60 villages in Mtwara in 2021 and handed over to randomly selected households. For most households, the transfer into the new homes was

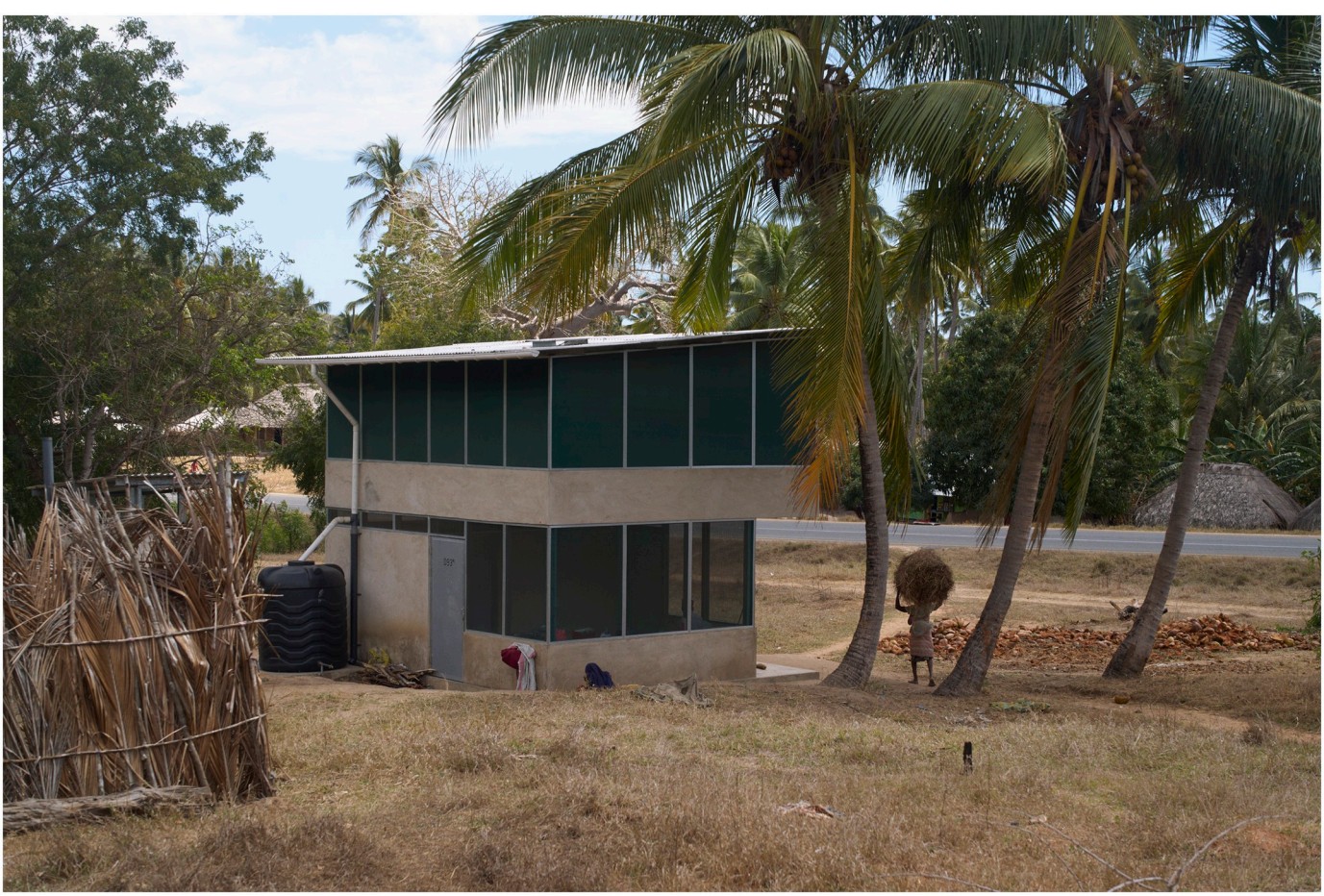

**Fig 1. A Star Home residents.**

uneventful, although a subset of households was reticent to move into new dwellings [30]. Questions remain how these new houses are used by their occupants and the response of the wider community to the longer-term presence of Star Homes. Drawing on several qualitative research methods, this article examines the response among residents of Star Homes and the wider community to the homes. We examine how Star Homes residents make use of the spaces and design elements and attitudes to the houses. The overall aim is to inform future iterations of such houses, with a view to ensuring their long-term use.

## Methods

### Study design

This descriptive, qualitative study incorporated a combination of methods that included observations, focus group discussions, and in-depth interviews over several phases of data collection (S1 Table). The study was designed by a multidisciplinary team, including social scientists, public health specialists, entomologists, and architects. The data were collected by a Tanzanian social scientist with more than 10 years of experience in qualitative methods (JM) with the assistance of Tanzanian field researchers. The data were analyzed by a team of experienced social scientists with backgrounds in low- and high-income settings (BA, JM, and CP) with input from the wider Star Homes team.

### Study setting

Mtwara is one of Tanzania's 31 administrative regions. The Makonde and Makua are the region's dominant ethnic groups and the region's economy is based mostly on farming. Around 85% of the land in Mtwara Region is arable, yet only about a quarter of this is used for cultivation. Livestock rearing is not popular and its contribution to the economy is negligible. Cassava, millet, and sorghum are important food crops. Cashew nuts are the most important cash crops. One fourth of the region's gross domestic product comes from cashew nut farming and makes up 40% of the country's total cashew nut production. Study communities were spread across the district. Eligible households within a village cluster were offered the opportunity to win a Star Home in a fair and transparent lottery. This was the first project of its kind in the region where few health-related research or development initiatives had been implemented previously.

The design and development of the Star Homes was informed by the results of the pilot project in Magoda [11]. The Star Homes' design was implemented by Danish architecture firm Ingvartsen (http://ingvartsen.dk/) in consultation with clinicians, entomologists, public health practitioners and social scientists. The light gauge steel structures were constructed by the Dar es Salaam based engineering firm Ecohomes (https://ecohomes.co.tz/). Where possible, novel elements were tested at 1:1 for occupant feedback as part of an iterative design process prior to the commencement of construction for the trial. This included three full houses and two latrine prototypes with different design, construction methods, and materials.

The health interventions of a two-storey Star Home are illustrated in Fig 2. The house and ventilated pit latrine design were developed around a lightweight steel frame, with large openings and a mono pitch roof. The openings are screened with shade net reinforced by wire mesh. The spatial layout was informed by a study of vernacular houses in the region and includes a screened kitchen living area, and first floor bedrooms, accessed through a locked store [31]. The Star Homes were designed for a rural context and function off-grid. They can be constructed and maintained with components available locally in Tanzania, including solar panels and rainwater harvesting.

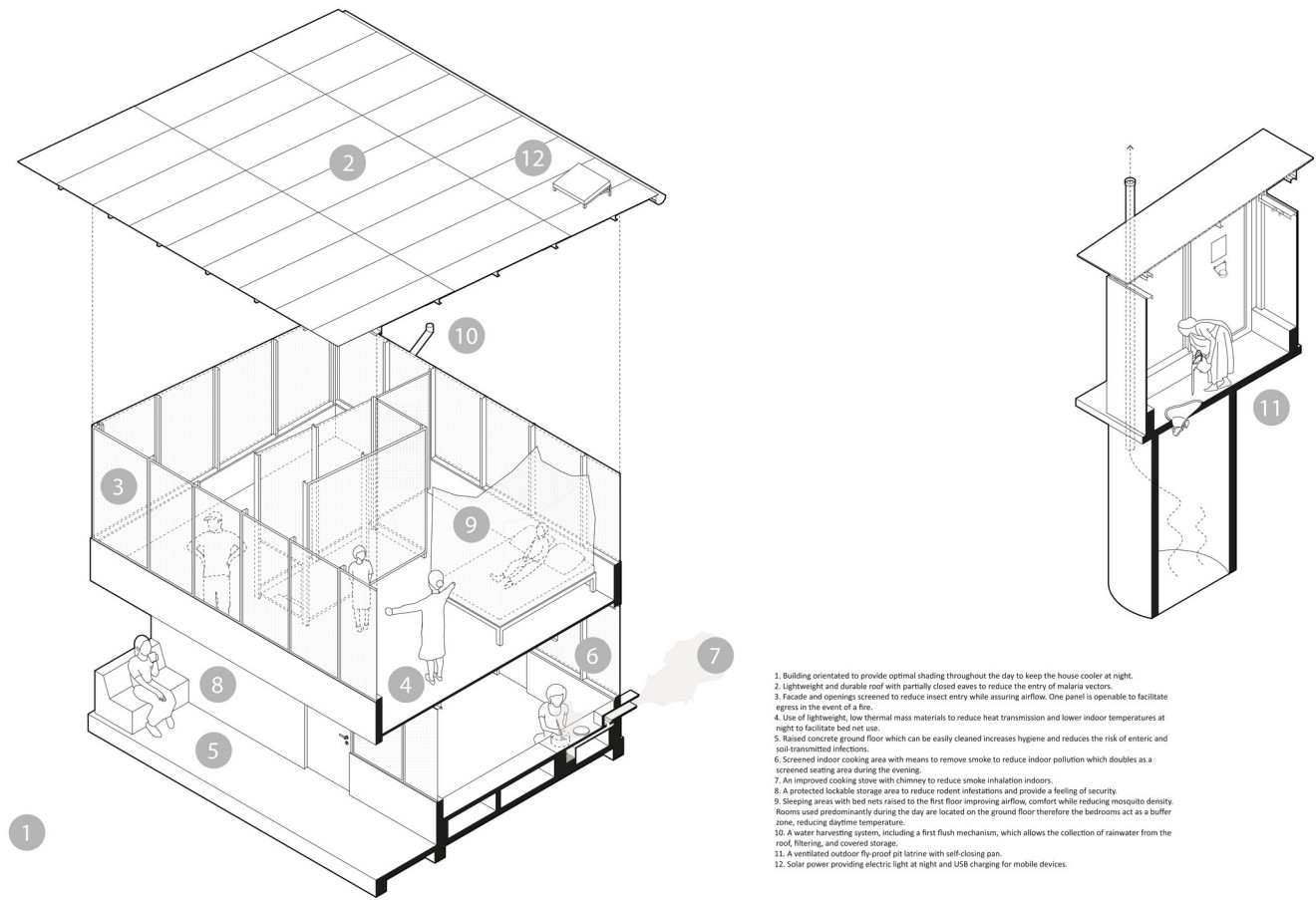

1. Building orientated to provide optimal shading throughout the day to keep the house cooler at night.
2. Lightweight and durable roof with partially closed eaves to reduce the entry of malaria vectors.
3. Facade and openings screened to reduce insect entry while assuring airflow. One panel is openable to facilitate egress in the event of a fire.
4. Use of lightweight, low thermal mass materials to reduce heat transmission and lower indoor temperatures at night to facilitate bed net use.
5. Raised concrete ground floor which can be easily cleaned increases hygiene and reduces the risk of enteric and soil-transmitted infections.
6. Screened indoor cooking area with means to remove smoke to reduce indoor pollution which doubles as a screened seating area during the evening.
7. An improved cooking stove with chimney to reduce smoke inhalation indoors.
8. A protected lockable storage area to reduce rodent infestations and provide a feeling of security.
9. Sleeping areas with bed nets raised to the first floor improving airflow, comfort while reducing mosquito density. Rooms used predominantly during the day are located on the ground floor therefore the bedrooms act as a buffer zone, reducing daytime temperature.
10. A water harvesting system, including a first flush mechanism, which allows the collection of rainwater from the roof, filtering, and covered storage.
11. A ventilated outdoor fly-proof pit latrine with self-closing pan.
12. Solar power providing electric light at night and USB charging for mobile devices.

**Fig 2. Essential components of a Star Home.**

Most houses in the study villages are wattle and daub constructions (aka mud huts or traditional homes), with earth floors, and a thatched roof. These houses lack piped water and electricity. A total of 110 Star Homes were built and are compared with 440 wattle/daub homes. A dynamic cohort of children was established whereby at least two children under 13 years per each house were enrolled for clinical surveillance, and any new child in the age range were enrolled afterwards for three years [22].

## Sampling

Residents of Star Homes and traditional homes were recruited for the current qualitative study because of issues they had reported to the research team who regularly visited the homes. Data collection occurred at approximately six-monthly intervals to capture changing perspectives as residents became accustomed to the Star Homes. Respondents were initially recruited by a research assistant with the help of District Malaria Focal Person who is Government employee and coordinates malaria control and prevention activities. Study respondents were familiar with the interviewer because of her previous interactions and was independently conducting the interviews and discussions guided by the research questions. The interviews were followed by observations and informal interactions with a total of four Star Homes to document the diverse spectrum of experience of living in the Star Homes. These four Star Homes were purposively selected based on the distinctive issues and problems faced by the residents.

## Data collection

In depth interviews (IDIs), focus group discussions (FGDs) and observations were conducted with residents of study homes using interview topic guides adapted from the previous round of interviews that explored the residents' reticence to enter Star Homes (S2 Table) [30]. The guides were modified to explore the use of the Star Homes, likes and dislikes of the structural components of the houses, overall living experience and adapted behavior when living in the houses. Respondents were visited in their houses for face-to-face interviews. Each interview lasted between 45 minutes to 1 hour. In addition, follow-up interviews and observations (including four case studies based on the discussions with residents) were carried out during November to explore the details related to their experience of living in Star Homes. The number of respondents for the interviews were guided by the principle of data saturation that meant no new information was generated after certain number of interviews [32].

After three rounds of data collection (round I: Oct-Nov 2021; round II: Apr-May 2022; round III: Oct-Nov 2022), a total of 43 IDIs and four FGDs were conducted with heads of households (Table 1). These included 33 IDIs with heads of Star Home, four IDIs with heads of wattle/daub control homes and six IDIs with heads of neighboring households outside the study cohort. The four focus group discussions were conducted with 30 participants consisting of 13 heads of wattle/daub control homes, and 17 heads of community members' homes outside the study cohort. The last round of interviews (round IV: Apr-May 2023) included eight interviews (four Star Homes and four control traditional home residents).

## Data management and analysis

Interviews and discussions were conducted in Swahili or Makonde and were audio-recorded. Audio-recorded conversations were transcribed into English for thematic analysis. Transcripts were read line by line and coded in QSR NVivo based on the codebook that was developed for the initial interviews and was adapted for the data. The codebook was modified to respond to research questions related to the use of Star Homes, challenges, and opportunities. Codes were coalesced to form themes supported by quotes (S3 Table).

## Ethical approval

The study was approved by the National Research Ethics Committee of Tanzania on 17 June 2021 (reference NIMR/HQ/R.8a/Vol.IX/3695) and the Oxford Tropical Research Ethics Committee on 2 July 2020 (reference #533–20). All respondents provided written informed consent to participate in the study (S4 Table). The individual pictured in Fig 3 has provided written informed consent (as outlined in PLOS consent form) to publish their image alongside the manuscript.

## Results

### Overview

Star Home residents described several benefits of the new homes, including more comfortable living conditions, better ventilation, water and sanitation systems, and lighter rooms. They also reported issues related to several design elements of the houses. For instance, the ingress of rainwater, uncomfortably cold floors during the colder months, inability to keep door closed, and the need for continued use of ITNs. There were issues related to the functionality of the stove (e.g. stoves required more time to cook certain foods, burnt more fuel, and produced inadequate heat). Residents therefore continued to use their traditional houses and/or outside spaces for cooking and eating. Some residents with small children preferred to sleep

**Table 1. Socio-demographics of respondents who participate in the study at various time periods.**

| | Round 1 (Oct-Nov 2021) | Round 2 (Apr-May 2022) | Round 3 (Oct-Nov 2022) | Round 4 (Apr-May 2023) | Total | % |
|---|---|---|---|---|---|---|
| **Gender** | | | | | | |
| Female | 15 | 7 | 7 | 4 | 33 | 38% |
| Male | 31 | 13 | 6 | 4 | 54 | 62% |
| **Total** | **46** | **20** | **13** | **8** | **87** | **100%** |
| **Age** | | | | | | |
| 18–29 | 1 | 0 | 0 | 1 | 2 | 2% |
| 30–39 | 9 | 3 | 5 | 2 | 19 | 22% |
| 40–49 | 14 | 7 | 5 | 2 | 28 | 32% |
| 50–59 | 11 | 7 | 1 | 0 | 19 | 22% |
| 60–69 | 7 | 2 | 1 | 3 | 13 | 15% |
| 70+ | 4 | 1 | 1 | 0 | 6 | 7% |
| **Sub-total** | **46** | **20** | **13** | **8** | **87** | **100%** |
| **Education** | | | | | | |
| College | 3 | 0 | 0 | 0 | 3 | 3% |
| Secondary | 4 | 0 | 0 | 0 | 4 | 5% |
| Primary (Complete) | 25 | 13 | 4 | 2 | 44 | 51% |
| Primary (Incomplete) | 1 | 1 | 3 | 2 | 7 | 8% |
| Not Attended | 13 | 6 | 6 | 4 | 29 | 33% |
| **Sub-total** | **46** | **20** | **13** | **8** | **87** | **100%** |
| **Livelihood** | | | | | | |
| Small farmer | 42 | 18 | 12 | 8 | 80 | 92% |
| Small farmer and Other | 1 | 2 | 1 | 0 | 4 | 5% |
| Other | 3 | 0 | 0 | 0 | 3 | 3% |
| **Sub-total** | **46** | **20** | **13** | **8** | **87** | **100%** |
| **Category** | | | | | | |
| Star Households | 10 | 10 | 13 | 4 | 37 | 42% |
| Comparison Households | 13 | 4 | 0 | 4 | 21 | 24% |
| Neighboring Households | 0 | 6 | 0 | 0 | 6 | 7% |
| Community members | 17 | 0 | 0 | 0 | 17 | 20% |
| Community Leaders | 6 | 0 | 0 | 0 | 6 | 7% |
| **Sub-total** | **46** | **20** | **13** | **8** | **87** | **100%** |

downstairs because it was closer to the toilet, and the difficulties that young children faced when trying to climb down the stairs on their own. In some cases, winning a new house also triggered jealousy within the family members and among other community members, which affected their relationship by inciting arguments, violence, and resentment.

## Overall comfort of Star Homes

The main benefit of the Star Home was the comfortable living conditions for the entire family (Table 2). Respondents spontaneously and favorably compared the comfort of Star Homes to that of the wattle/daub houses. For example, they reflected on the need to collect grass for seasonal roofing, trying to keep the earth floors clean, and the time and energy that these activities required. Furthermore, residents explained that living in wattle/daub house required cleaning the bugs, dirt, dust, and sometimes fallen roofing materials and reptiles. In contrast, Star Homes required no such additional labor and resources for cleanliness. The overall structure

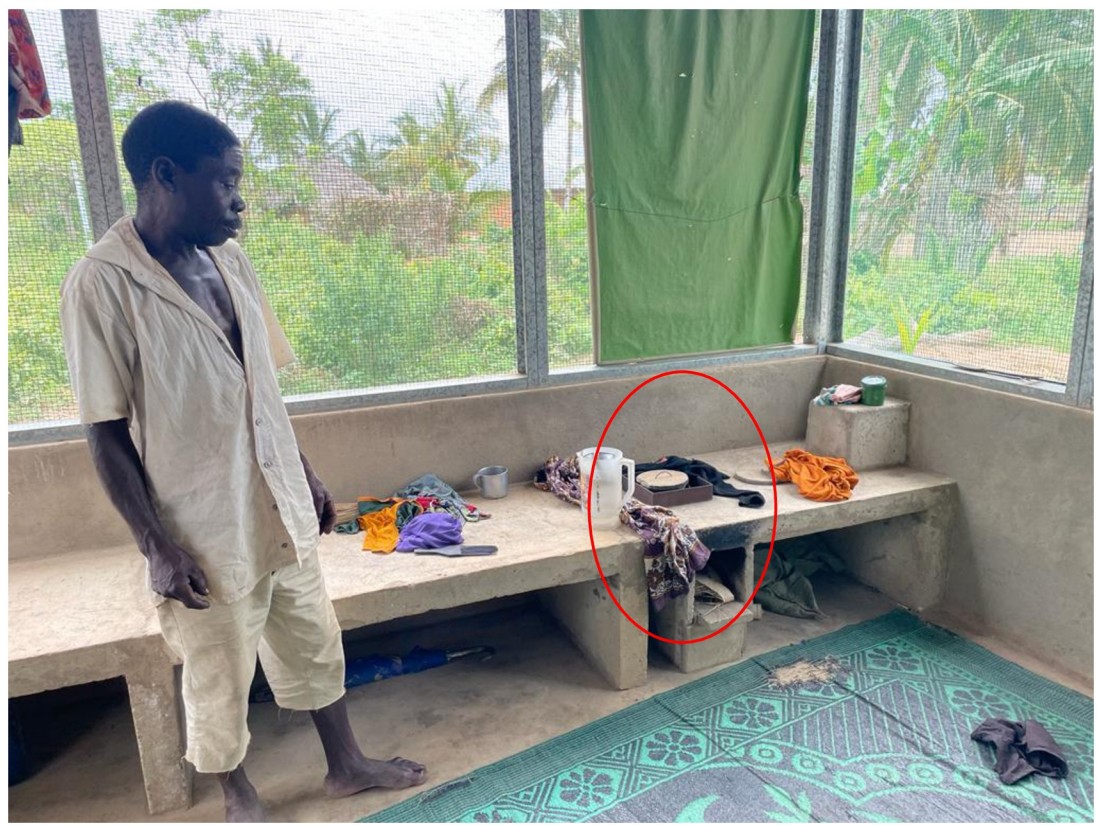

**Fig 3. A Star home resident showed his stove and a shield (marked by a red ring) on the shade net.** The issue of water ingress on the ground floor was also raised by residents. This was despite occasional holes in the shade nets being fixed by the Star Homes team. Sometimes, the stove fire was extinguished by the rainwater coming into the kitchen.

of the house, concrete floors, the access to a reliable and close source of water, and the fact that the living space was distributed across two floors added to the perceived comfort and was positively contrasted with the crowded living conditions in their previous homes. Respondents emphasized that this comfortable living space had been provided free of charge.

## Entering and leaving the dwelling

Residents were accustomed to unrestricted movement between the space inside and outside wattle/daub houses. Many wattle/daub houses had no lockable or closable doors and, when present, residents were observed rarely closing the door, especially during the daytime. Thus, going along with the request of the Star Homes project staff to keep the doors closed as much as possible—to reduce the ingress of mosquitos—required a change in their daily practices. Aware of this issue, the Star Homes were fitted with sprung, self-closing doors. At times, these doors were observed propped open (Table 3). Moreover, their hinges and locks required frequent repair, which was undertaken by the project and took longer than hoped. Hence the Star Home doors sometimes had to be closed manually. And respondents reported that their children were incapable of keeping doors closed, and left doors open despite their constant reminders—indeed, parents were aware that mosquitoes entered the house when the door was left open. Furthermore, when parents were away at their farms, children remained in the house and door closure was not supervised.

**Table 2. Case study I.**

| Case study I | **Family background:** A 59-year-old man who has been living in a Star Home with his wife (the head of the household) since 2021. They have three children, 15, 11 and 8 years old. He is a small-scale farmer who cultivates cassava, rice, and maize. |
|---|---|
| | **Use of upstairs:** During a walk-through interview and observation it was found that the Star Home residents were actively using and living in the house and they slept in the rooms upstairs. The parents' sleeping area was located on the right and children's sleeping area was on the left. All three children slept in one room with different sleeping spaces. The bed nets seemed to be in use and the resident reported that the ventilation upstairs was good; not too hot not too cold. The household head liked the Star Home because sleeping was comfortable even when they forgot to sleep under the bed nets, there were no mosquitoes but during windy rains, a lot of water entered the house through the facade and it became inhabitable. |
| | **Use of downstairs and living area:** The household members used the room downstairs (the kitchen) as a sitting area when they had visitors usually between 6pm and 7pm, but it was not conducive during the rainy season as water entered into the area through the shadenet facade. If they could afford it, they would improve it by adding half a wall around the area. |
| | **Use of storage:** They used the storage area for keeping food and water buckets, also utensils, and other equipment. |
| | **Use of the stove:** This household reported using the Star Home stove for all their cooking because it was the only cooking stove they had and it worked well except when it rained, then the kitchen area used to get flooded. If it rained it was not possible for them to cook. They would go to bed hungry. He also reported that they always ate in indoor areas to conform to project requirement that eating outside may expose them to mosquito bites and possible food contamination. |
| | **Children evening routines:** Most of the times children entered in the house between 7:30 pm and 8 pm for dinner and thereafter they went to sleep. This area being near the border to Mozambique, there were seasons when people were not allowed to stay outside after 8pm for security purposes. |
| | **Toilet use:** The toilet facility worked well. He and his family used it comfortably and did not have any challenges in contrast to their (previous) traditional house, the toilet used to be eroded every now and then during the rainy season(s) where they had to rebuild it repeatedly. |
| | **Use of water:** The water system worked well with minor challenges on the water tap but this family was happy to have access to clean water throughout the rainy seasons. |
| | **Star House doors:** He felt like the Star Home door locks were not child friendly because they tended to break apart when children handled them. |
| | **Other issues:** This household experienced issues related to envy from some community members who did not participate in the Star Home project. Some community members condemned them as not deserving to live in such a beautiful house. |

*I: What do you think are the reasons that led to frequently unclosed doors into this house?*

R: *Children*

IDI3 HAC MAG 042_270423

## Sleeping inside

As part of the study, the recipients of Star Homes, were asked to sleep upstairs, and under the bed nets. Interviews with Star Homes household heads, however, found that this was not always the case, for several reasons, including: (1) the shade-netting in the upstairs windows leaking badly during heavy rains; (2) the transparency of the shade netting raising concerns about privacy; (3) the upstairs space was described as cold during the cooler months; (4) fears among parents and children of using the stairs at night because the lights in this space were not always operational (affected by the ingress of water, limited solar charging and their extended using during nighttime period). The Star Homes project team was observed to be responsive to such concerns, for instance, providing curtains, blankets, and mattresses.

Another reason why sleeping upstairs was not popular among the residents, particularly young children, and pregnant woman was the difficulties that children and residents experienced in descending the stairs when going to the toilet during the night (Table 4). Children

**Table 3. Case study II.**

| Case study II | **Family background:** A 30-year-old female head of Star Home (household) earns her living by farming. She lived with her husband and four children including three boys (8 and 4 years old and one 8 months old) and a 10 years old girl. They returned once to their previous house when the Star Home caught fire from an accidental explosion of a Jerry can containing petrol placed close to an open fire. Later, the Star home was renovated by the project team and the residents returned to their new home.<br>**Use of upstairs:** The whole family (husband, wife and their four children) were using rooms upstairs to sleep. Although they slept upstairs, all children experienced repeated episodes of malaria. Earlier in the study, children complained that sleeping upstairs, on the floor, without mattresses was very cold, and they preferred to sleep downstairs in the storage area where it was warmer. But after being provided with mattresses and blankets, they started sleeping in the rooms upstairs.<br>**Use of the storage area/room:** The family used the storage area to store various materials and crops. Her children (7 and 9-years-old) were alternating between sleeping upstairs and downstairs. The woman also slept upstairs and downstairs. For example, during her pregnancy, she slept upstairs throughout the pregnancy but after delivery she could not stay upstairs because she had to climb the stairs. So, she stayed and slept downstairs in the storage room area for about three weeks.<br>**Use of the kitchen/stove:** This Star Home's stove had cracks even before the house caught fire. The stove had deteriorated to become unusable, so the family used a kitchen in their old house, but they normally moved to eat in the Star Home either outdoor or indoor because there was light in a Star Home. Before the house got burnt, they used the Star Home stove to cook some of the foods but when it came to cooking cassava stiff porridge the Star Home stove could not handle the large pot. The other challenge was when it rained, the rainwater entered through the facade, the area downstairs got flooded to an extent that it compelled her to wish and afford that she could renovate the place by building a wall to block rainwater.<br>**Evening routines:** The family normally stayed outdoors at the veranda from around 8pm to 10pm that consisted of the mother, father, and the 8 months old baby.<br>**Use of toilet:** She liked the toilet facility, and the family used it without facing any problem.<br>**Use of water:** She expressed her appreciation for having the water tank since the village had challenges related to water. Having water source at her dwelling was an invaluable benefit.<br>**Star Home doors:** Keeping the doors closed all the time was a challenge to this family since children could not ensure the door closure all the time when no adult was around. The door could stay opened for two to three hours in the evening around 6pm and 7pm.<br>**Other issues:** She liked the Star Home because even when it rained there was not so much rainwater inside the house compared to the old house where the roof was leaking during the rainy season. She reported that Star Home roof also leaked through the ceiling where the light bulb was inserted. Because the Star Home caught fire, residents returned to their old house temporarily until their Star Home could be repaired. Returning to their previous home they experienced the worst of dust, fleas, and mice. When it rained, the old house roof also leaked to an extent that they woke-up to find their house full of water which really hurt them. But overall, the relative benefits of Star Home were incomparable to the old house. Referring to the Star Home, the resident said, "*this house is good, water is available, the lights work well, we sleep on a nice place, in short it has no problems at all*". Since she also had a small business selling sardines and tomatoes, with the Star Home, she can keep/store her sardines safe from mice, houseflies and or chicken that otherwise would destroy them in the old house. But when the Star Home was burnt, her stored goods all got burnt and she could not yet find the capital to resume her business. |
|---|---|

urinating upstairs was an issue for some families because the concrete bedroom floors—in contrast to earth floors—do not absorb urine.

*I: Why do you sleep downstairs with infant?*

*R: Because of going outside now and then, to take care of young child is challenging sometimes may poo, you need water to clean so she will need to go down to throw feces in latrine, that is why we decided to sleep downstairs*

IDI8_ASN_NJU_036_081122 (Results-III)

**Table 4. Case study III.**

| Case study III | **Family background:** A 49-year-old female head of a Star Home earns her living through farming activities. She moved into the Star Home together with her five children and grandchildren the same day she was handed the house keys in 2021. She and the children have been living in the Star Home since then and they liked most components of the house, for example the benefits of having toilet close to the house. Nonetheless, there were few challenges too that she shared with us, that was mostly the rainwater entering the house during rainy season which made it impossible to continue normal activities. |
|---|---|
| | **Use of upstairs:** A four-year-old child slept downstairs with the grandmother while the other three children slept upstairs. She wanted the emergency window located upstairs to be locked because a thief entered the house through that window via the water tank and stole several possessions of the family. |
| | **Use of storage:** The wife, husband and a four-year-old grandchild slept downstairs at the storage area, despite that they were aware that the space was meant for storage. The family did not have a lot of crops to occupy the area so there was enough space for sleeping. About three sacks of crops were seen on one side of the storage room adjacent to the beds. She slept downstairs with her four years old grandchild especially because it was easier for them to use the toilet at night. |
| | **Use of kitchen/stove:** She initially used the Star Home stove, but it got cracked along the way, so the stove was no longer in use. The family was found cooking on a charcoal stove outside near the main entrance to the Star Home. She was also using an old house as a kitchen and reared chicken and cattle there. |
| | **Use of toilet:** The toilet was in good condition, and the family used it all the time with only minor problems such as a broken ventilation pipe that had a small hole but it did not compromise their use of the toilet. |
| | **Use of water:** The family used and appreciated water from the Star Home tank, but the tank's lid had some problems such that when it rained the lid flipped away and water did not collect into the tank. |
| | **Other issues:** When it rained, the Star Home windows allowed entry of rainwater and dust into the house, and that made the residents disappointed although she liked the house in all other aspects. One of her favorite spots was the veranda near the door, where all family members gathered for dinner. And sometimes, they simply ate their cooked food inside the house. |

## Water, sanitation, and cooking

Each Star Home had a 2000L water tank, placed adjacent to the house. This water tank collected rainwater from the roof, which had been specially designed with a first flush system to minimize debris from the roof entering into the tank. Residents were able to drain the water tank through an outlet tap and were seen collecting water in the bucket and/or large basin for washing vegetables, fish, meat, clothes and utensils after cooking. The water tank and its large capacity was a point of particular praise among Star Homes residents: they were impressed by the durability of the water collected in the tank that allowed them to use water for a wide variety of activities without having to replenish it. Star Homes residents appreciated the ventilated pit latrines, which were located close to the house. Some explained their preference for the new toilets because of the ease of access.

Even though Star Homes were fitted with a specifically designed stove (Fig 3), integrated into the indoor cooking area, some residents described cooking food outside of the home or in their previous traditional house which were not completely abandoned. Foods that required prolonged heat, such as cassava stiff porridge, especially when cooking for a large family size, were said to be cooked on traditional stoves but were dined at Star Homes (Tables 3 and 4). Switching the stoves between Star Homes and traditional homes was common because of the concerns residents shared with us. Respondents suggested that stoves in Star Homes consume more fuel, and produces less heat than their traditional way of cooking. The cooking stoves were therefore not popular among the residents.

*I: And when you cook, you are cooking in the old house.*

*R: Yes, we cook there because cooking cassava ugali is difficult.*

*I: Yes, I say that's why you cook there and eat there*

*R: No, we cook there and eat here*

IDI4_SAM_MANGO_038_041122

Upon further probing the reasons for the underuse of stoves, residents described additional slippery metal brackets used to hold the pots for cooking, the small size of the stove's burner, and the long distance between the firewood and the potholder, the latter reportedly meant that more firewood was consumed. Some described the impact of wind and rainwater on the burning of the firewood, which led them to add a shield on the shadenet wall.

The issue of water ingress on the ground floor was also raised by residents. This was despite occasional holes in the shade nets being fixed by the Star Homes team. Sometimes, the stove fire was extinguished by the rainwater coming into the kitchen.

R: *When it rains, all the rainwater pours in the stove place and the fire was put out.*

I: *But when it happens like that, the fire always goes out*?

R: *Yes, because of the water.*

I: *Aaaah*

R: *When there is heavy rain; you can find a lot of water there.*

I: *Aaah!*

R: *The downstairs will be full of water due to the heavy rain.*

IDI2 HAK2 MAK-A_031_260423

## Response of the wider community

The responses of the wider communities to the Star Homes (and their residents) were mixed. Star Homes and control home residents described comments by their neighbors that indicated jealousy on their part. Neighbors were jealous because the study participants were perceived to be lucky to have been selected to participate in the study and benefit from the free health care that came with it, regular health checks and gifts. In contrast to the early phase of the study, when neighbors made malicious comments about winning star homes and staying in the houses, in later months, they offered positive comments and feedback. During daily interactions with their neighbors, residents were reminded that they were '*lucky*' and '*blessed*' to be study participants (Table 5). Neighbors also perceived research staff to have encroached into their private spaces in off hours, particularly referring to their early morning visits as a part of the research procedure. Neighbors often alleged research staff as bad people.

*I: Your neighbor?*

*R: Yes, the neighbors, . . . . . . some of them they said that we're being reared like local chickens.*

*I: Being reared like a chicken?*

*R: Yes, being reared like a chicken but receiving health services on time.*

IDI1 ASN MAK-A 201

**Table 5. Case study IV.**

| Case study IV | **Family background:** A 43-year-old female head of a Star Home was a small-scale businesswoman who also engages herself in farming activities. She lived with her 13-year-old son and a 10-year-old grandson. After being offered a Star Home, the grandson initially did not want to live with her, he used to refuse his blood examination when project staff visited the Star Home. Part of the reasons for refusal was because his parents did not approve of the study and its procedures fully. Later, the parents agreed, and the grandson also participated in the study. |
| --- | --- |
| | **Use of Upstairs:** The boys slept in one of the rooms upstairs, the other room where the solar charging system was placed was unoccupied, usually utilized when the household received visitors to sleep overnight. The boys slept in one room, the room the resident showed did not have bed nets. When asked, she reported that the bed net was removed to be washed the same day. The bed net was hanging on a rope to dry outside and was in a good condition. |
| | **Use of downstairs living room area:** When she had visitors at the household, they normally sat wherever they found her either indoor or outdoor on the veranda. If the visitors came around 8pm or 9pm they usually met indoor, in the living room area near the kitchen/stove. The reason to stay inside at night hours was mostly because there were too many mosquitoes outside although she reported that there were less mosquitoes during the dry season. |
| | **Use of storage room downstairs:** She was aware that the storage area was a living room, but she used it as a sleeping area. She did not have anything to store but when she harvests she will be storing the harvest upstairs. This led to her dilemma whether she should be using the downstairs areas for storage or for sleeping. |
| | **Use of kitchen/stove:** The Star Home's stove was used only when the grandmother was cooking but when the boys cooked they used an outdoor kitchen in her old house because the Star Home stove had cracks and was delicate. She believed the boys were not able to handle it with care thus she did not allow them to use it. |
| | **Children's evening routines:** The children entered the Star Home around 7:30pm because they were normally alone at home and was a time that she returned. When they were scared to stay outdoors by themselves, they locked the doors and stayed indoors. |
| | **Star Home doors:** The Star Home doors were always closed except when someone was visiting or leaving the house. She explained that the only door that remained open most of the time was the toilet door. The locks of the toilet door were out of order and were no longer working at the time of the visit. |
| | **Use of toilet:** She liked the toilet being permanent. The only challenge was a broken lock otherwise the toilet worked well, and she was happy to use it. |
| | **Use of water:** She liked the fact that rainwater was available in the Star Home so that she did not have to go to fetch water around the community from wells. The water collection pipe to the tank was once broken but it was later fixed by her elder son and has been in continuous use since then. |
| | **Other issues:** She liked the fact that she and her grandsons were no longer getting malaria episodes because living in a star house they were protected from mosquito bites '*the repeated episodes of malaria have no longer happened to me and my sons. . .as you know nowadays, we are protected*' but she did not like rainwater entering the house, mattresses were soaked and were unusable. |
| | The neighbors and people around in the community were currently congratulating her that she won the house '*congratulations to you for getting a good house where you live.*' They now wished to have also won a Star Home. They have changed their views and attitudes towards the Star Homes project because they have entered the Star Home. Nowadays, when it rained, neighbors came to collect rainwater from the Star Home tanks. |

Star Homes residents were asked if they were stigmatized or discriminated against by neighbors just because they won a Star Home. Some respondents reported neighbors made comments that the Star Home residents were living on aid and made negative comments such as the houses were only for Christians. However, the residents indicated that many of these comments had subsided after living in the house for some months.

## Discussion

Participants in the Star Home study appreciated the opportunity to participate in the project. Star Home residents particularly valued their features and comfort they offered, such as water, lights, ventilation, and toilet. Nonetheless, there were issues related to several aspects of the house, such as rainwater ingress, functionality of the stove, and smoke production. Because of the sampling approach, with the over sampling of households that reported problems, this is

likely an over-reporting of issues with the design elements. Star Homes residents and controls residing in waddle/daub homes received both compliments and jealousy from their neighbors.

## Wattle/daub versus Star Homes

Star Home residents emphasized the benefits of the house in terms of comfort and additional features. For instance, the ventilated spaces, water storage system and latrine, which were absent in wattle/daub houses. Wattle/daub houses are often dark, minimally ventilated, grubby, and invariably single storied building [25], although there is diversity among the more conventional houses seen in rural (peri-urban and urban) sub-Saharan Africa [33]. Elsewhere in Tanzania, researchers have highlighted the need to improve housing to facilitate malaria control, pointing to the limitations of existing structures, such as broken windows, holes in the walls, door covers, eave spaces, and roofs [34]. A strong preference for improved housing in contrast to traditional housing was also found in The Gambia [35].

Although living in Star Homes was considered a privilege, components of the house required maintenance and repair for which the residents relied on project staff. The residents were advised not to attempt repairs and to contact the project, although this sometimes meant that repairs took longer than desired. Based on the willingness to learn new skills within the community, building capacity to maintain houses would be a shrewd solution to mitigate the waiting time and resources required for repairs. Building a maintenance culture through participatory methods, including co-learning, and skills transfer aligns with the principles and elements of community engagement [36–38] and can engender independence and sustainability of the home [39]. Although the breakdown of building elements (e.g. door locks, shade nets, cooking stoves, light bulbs, and water tanks) were minor problems, their impacts on daily living were high. In a longitudinal study in Gambia that examined the durability and functionality of house modifications designed to reduce the entry of malaria mosquitoes, the major issues were found around the door and its use, that included poorly functioning door locks, handles and loss of automatic closure mechanism [40]. In future, maintenance and repairing mechanisms should be resourced within the community to shorten the repair cycle.

## Internal and external spaces

Star Homes are intended to protect against malaria through preventing vectors entering the structure, particularly the sleeping area. The façade openings are screened with shadenet, the sleeping area is located upstairs and, when closed, internal and external (self-closing) doors are mosquito proof. The results indicate that the protection against mosquito bites was compromised when the residents did not sleep upstairs, did not sleep under the ITNs, or keep the doors open. This was despite sensitization and instructions from the Star Homes team, and the provision of ITNs at the start of the study. The finding underlines the need for multi-disciplinary research on housing interventions that incorporates analysis of residents' use of spaces. Such research can potentially contribute to explaining the impact (or lack thereof) of housing interventions on health outcomes, such as malaria, respiratory tract infections and diarrheal diseases and is currently being evaluated by the clinical study [22, 29].

The tendency not to sleep upstairs or under ITNs highlights how human behavior remains a central determinant in the effectiveness of malaria prevention interventions [41]. The finding reflects a wider trend of sub-optimal ITN use: research in Tanzania has indicated that only 55% of under five children sleep under ITNs [42]. In the Star Homes Project, gifts and incentives together with messages on the health benefits seemed to motivate children and parents to sleep under bed nets. Such an approach was necessary to evaluate the impact of Star Homes when used as intended. However, this is unlikely to be sustainable outside of study conditions.

The findings highlight how changes in daily practices—for example, sleeping location—can be highly contingent on design details (such as staircase lights and cold urine-resistant concrete floors). Under appropriate conditions, changes occur.

## Envy

As a result of its study design, the communities where the Star Home project was conducted were divided into three categories: Star Home residents (intervention households); residents of the control wattle/daub homes, and other community members. Beyond providing a new dwelling to the intervention households, participation in the Star Home project also entailed additional benefits, including health insurance and incentives to sleep upstairs and use the ITNs. These benefits—combined with the new homes—triggered accusations towards the project staff and participants, which indicated envy in the wider community, particularly during the early phase of the project [30]. Envy stems from the desire to acquire something possessed by another person (as opposed to jealousy, which is rooted in the fear of losing something already possessed) [43]. Envy is often expressed to third persons (i.e. neighbors/community members) through gossips, backbiting, criticisms, as a potential tool to dissuade people who have achieved the wealth, material or status, more than theirs [43]. In the months after construction was completed, Star Homes study staff were accused of being 'freemasons' and of malicious intentions towards the participants; participating families were reproached for putting their children in harm's way [28, 30]. However, the interviews conducted after a year of residency indicated that, although respondents still described some persisting envy, the ill feeling had subsided. The response of the wider community echoed that of other randomized controlled trials in which participants are offered ancillary healthcare (whether in the control or intervention group) [44–49]. Given the scale of benefits that participants received i.e. a new house at least in the initial stage the jealousies were even greater than those that can accompany a more conventional clinical trial in poor communities.

The Star Homes project employed commonly used strategies to address the accusations: conducting qualitative research to understand the "rumours", and undertaking a responsive program of community engagement, including football and netball tournaments to disseminate information about the project and to humanize project staff [50, 51]. Future large-scale community-based vector control interventions, particularly those focused on housing, can learn from this study and ensure that there are appropriate mechanisms for community listening, as part of an adaptive and tailored approach to community engagement.

## Strengths and limitations

In contrast to the cross-sectional exploration of an exposure to an event or an intervention, this study captured at several timepoints over years the lived experiences of the residents in Star Homes and the waddle/daub houses. Star Home residents' experience and interactions with the house, and those of the wider community, are dynamic and vary over time, and this was captured using a longitudinal approach. Collecting data at multiple time points allowed the study to adapt to emerging issues and to the need to inform the wider ongoing study of health impact. Another strength was the maximum diversity sampling approach, whereby Star Homes residents and members of various communities were interviewed, which included seeking outliers and extreme cases based on the suggestions from the project team. In terms of limitations: although the interviewer was a researcher (who came from outside the study team), study residents may have been affected by desirability bias, and thus the perspectives in this study may not necessarily represent their actual opinion. Nonetheless, respondents were

quite candid about their experiences and the interviewer often probed to explore their challenges of living in Star Homes.

## Conclusions

Residents appreciated living in Star Homes, mostly because of the comfort they offered as well as the water and sanitation system, lights, and ventilation. They raised concerns around some of the novel design elements and highlighted the need for regular repair and maintenance. In some houses, the potential health benefits of the novel designs were impacted by residents not sleeping upstairs, or under ITNs and leaving doors open. This highlights that, even with close attention to the building design and careful sensitization around the potential benefits of dwellings, it can be difficult to ensure the intended health impacts of interventions for all. With concerted efforts to incentive sleeping upstairs, changes in practices were observed. Although the response of the wider community was not always positive, particularly during the initial stages of the project, jealousies had seemingly eased over time. The jealousies and negative reactions were entwined with unease about the wider benefits of project participation and the poverty in which many community members lived.

## Supporting information

**S1 Table. COREQ: COnsolidated criteria for REporting Qualitative research.**
(PDF)

**S2 Table. Interview guide adapted for the study.**
(PDF)

**S3 Table. Codes, themes and quotes.**
(PDF)

**S4 Table. Inclusivity in Global Health.**
(PDF)

## Acknowledgments

We are grateful to the respondents who participated in the interviews and permitted the researchers to enter their homes. We would also like to thank the wider Star Homes team and the communities where the Star Homes were located.

## Author Contributions

**Conceptualization:** Salum Mshamu, Judith Meta, Bipin Adhikari, Jakob Knudsen, Jacqueline Deen, Lorenz von Seidlein, Christopher Pell.

**Data curation:** Salum Mshamu, Judith Meta, Bipin Adhikari, Salma Halifa, Arnold Mmbando, Christopher Pell.

**Formal analysis:** Judith Meta, Bipin Adhikari, Christopher Pell.

**Funding acquisition:** Lorenz von Seidlein.

**Investigation:** Salum Mshamu, Judith Meta, Salma Halifa, Arnold Mmbando, Hannah Sloan Wood, Otis Sloan Wood, Thomas Chevalier Bøjstrup, Nicholas P. J. Day, Steven W. Lindsay, Jakob Knudsen, Jacqueline Deen, Lorenz von Seidlein, Christopher Pell.

**Methodology:** Salum Mshamu, Judith Meta, Bipin Adhikari, Hannah Sloan Wood, Steven W. Lindsay, Lorenz von Seidlein, Christopher Pell.

**Project administration:** Salum Mshamu, Christopher Pell.

**Resources:** Salum Mshamu, Hannah Sloan Wood, Otis Sloan Wood, Thomas Chevalier Bøjstrup, Lorenz von Seidlein.

**Software:** Judith Meta.

**Supervision:** Salum Mshamu, Bipin Adhikari, Hannah Sloan Wood, Otis Sloan Wood, Nicholas P. J. Day, Steven W. Lindsay, Jakob Knudsen, Jacqueline Deen, Lorenz von Seidlein, Christopher Pell.

**Validation:** Salum Mshamu, Judith Meta, Lorenz von Seidlein, Christopher Pell.

**Writing – original draft:** Judith Meta, Bipin Adhikari.

**Writing – review & editing:** Salum Mshamu, Bipin Adhikari, Salma Halifa, Arnold Mmbando, Hannah Sloan Wood, Otis Sloan Wood, Thomas Chevalier Bøjstrup, Nicholas P. J. Day, Steven W. Lindsay, Jakob Knudsen, Jacqueline Deen, Lorenz von Seidlein, Christopher Pell.

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
