## [Decision Letter · Decision Letter 0]

14 May 2024

PONE-D-24-05994Community responses to a novel house design: a qualitative study of “Star homes” in Mtwara, southeastern TanzaniaPLOS ONE

Dear Dr. Adhikari,

Thank you for submitting your manuscript to PLOS ONE. After careful consideration, we feel that it has merit but does not fully meet PLOS ONE’s publication criteria as it currently stands. Therefore, we invite you to submit a revised version of the manuscript that addresses the points raised during the review process.

We look forward to receiving your revised manuscript.

Kind regards,

Yitagesu Habtu Aweke, Ph.D

Academic Editor

PLOS ONE

Journal Requirements:

 [This study was funded by Hanako foundation, Singapore.].  

4. We note that Figure(s) 1 and 3 in your submission contain copyrighted images. All PLOS content is published under the Creative Commons Attribution License (CC BY 4.0), which means that the manuscript, images, and Supporting Information files will be freely available online, and any third party is permitted to access, download, copy, distribute, and use these materials in any way, even commercially, with proper attribution. For more information, see our copyright guidelines: http://journals.plos.org/plosone/s/licenses-and-copyright.

a. You may seek permission from the original copyright holder of Figure(s) 1 and 3  to publish the content specifically under the CC BY 4.0 license.

5. Please upload a copy of Supplementary File 1, Supplementary File 2 and Supplementary File 3 to which you refer in your text on page 22. Please amend the file type to 'Supporting Information'. If the Supplementary file is no longer to be included as part of the submission please remove all reference to it within the text.

Additional Editor Comments:

Please briefly elaborate your argument about the possible link between your case as risk  & the  the mentioned diseases. If possible, following a theoretical framework may make sense of your findings.

Reviewers' comments:

Reviewer's Responses to Questions

**Comments to the Author**

1. Is the manuscript technically sound, and do the data support the conclusions?

Reviewer #1: Yes

Reviewer #2: Yes

2. Has the statistical analysis been performed appropriately and rigorously? 

Reviewer #1: Yes

Reviewer #2: N/A

3. Have the authors made all data underlying the findings in their manuscript fully available?

Reviewer #1: Yes

Reviewer #2: Yes

4. Is the manuscript presented in an intelligible fashion and written in standard English?

Reviewer #1: No

Reviewer #2: Yes

5. Review Comments to the Author

Reviewer #1: There are issues with how this article has been written, organised and structured, overall. I suggest going through PLOS ONE guidelines for structuring the article to begin with.

- Introduction section is too loaded with the information about why novel houses are needed in the community. This section will be more balanced with maximum 2 short paragraphs on their relevance in preventing mentioned diseases, and the rest on how in other similar contexts communities have responded to living in tailored houses.

-The sub sections in the Methods section needs to be titled respectively as Study Setting, Sampling, Data Collection, Data Management and Analysis and Ethical Considerations. The sub-section, Study Design seems an unnecessary addition. The description of who did what can easily fit into the suggested subtopics. Also, mentioning names of architect firms with their links in the bracket looked amateurish to me.

- I suggest not to provide an overview of Results and directly get to the point. The Overview anyway presents a description of collected data, which should fit into the Data Collection section.

- Illustration of case studies looked abrupt to me, as the article did not mention anywhere that individual cases were studied, and even if it had, there needed to be clear mentioning of sampling procedure for the selection of those cases.

- In some instances, the result section jumps to conclusion without formulating proper argument to back it up. Result section should not present any discussion or conclusion. For e.g, p17: "Some respondents reported neighbors made comments that the Star Home residents were living on aid and made negative comments such as the houses were only for Christians. Such comments were manifestations of jealousies." To assert someone's opinion as jealousy is not a very academic way of sharing findings. Authors could have said (if applicable), "Respondents said they sensed jealousy in their neighbor's comments", and back it up with a quote. Further, I would avoid discussing Jealousy as one of the topics in the Discussion. Please rephrase it as something similar to "Unfavorable comments from the neighbors". The neighborhood not getting Star Houses could have (reasonably) triggered certain reactions but labeling them as jealous people makes the findings look biased.

Reviewer #2: • Please explain how "Star Home" protects its residents from malaria, ARIs, and diarrheal infections.

• Mention the criteria for recruiting respondents.

• Figure 2 is unclear; please provide a clearer figure.

6. PLOS authors have the option to publish the peer review history of their article (what does this mean?). If published, this will include your full peer review and any attached files.

Reviewer #1: **Yes: **Bibhu Thapaliya

Reviewer #2: No

---

## [Author Response · Author response to Decision Letter 0]

18 Jun 2024

3rd June 2024

Dear Prof. Yitagesu Habtu Aweke and reviewers,

Thank you very much for your comments/suggestions and opportunity to revise our work. We have utilized your suggestions to improve our manuscript. Please see below our responses to your specific suggestions with corresponding changes in the manuscript. 

We look forward to your kind consideration. 

On behalf of all the authors,

Bipin Adhikari

COMMENTS AND SUGGESTIONS 

Journal Requirements:

AUTHORS: Thank you for the style guidelines. We have followed the style guidelines in our revision. 

 [This study was funded by Hanako foundation, Singapore.]. 

AUTHORS: Thank you for the suggestion. Revised as suggested. 

AUTHORS: Thank you for the suggestion. Data is available upon request to MORU’s data access committee. We have added the email address, and the revised statement reads as follows: 

Because of the nature of the qualitative data in this study, even if the data are anonymized, potential respondents are identifiable. Both MORU and local ethics committee restricts the sharing of data that can potentially identify the respondents. Nonetheless, the data is available upon request to the Mahidol Oxford Tropical Medicine Research Unit Data Access Committee (datasharing@tropmedres.ac) complying with the data access policy on case-by-case analysis (https://www.tropmedres.ac/units/moru-bangkok/bioethics-engagement/data-sharing/moru-tropical-network-policy-on-sharing-data-and-other-outputs). 

4. We note that Figure(s) 1 and 3 in your submission contain copyrighted images. All PLOS content is published under the Creative Commons Attribution License (CC BY 4.0), which means that the manuscript, images, and Supporting Information files will be freely available online, and any third party is permitted to access, download, copy, distribute, and use these materials in any way, even commercially, with proper attribution. For more information, see our copyright guidelines: http://journals.plos.org/plosone/s/licenses-and-copyright.

a. You may seek permission from the original copyright holder of Figure(s) 1 and 3 to publish the content specifically under the CC BY 4.0 license.

[ 

AUTHORS: Thank you for the suggestion. We have replaced figure 1 (the revised figure does not have identifiable individuals) and for figure 3, we have collected consent from the individual shown in the photograph. 

5. Please upload a copy of Supplementary File 1, Supplementary File 2 and Supplementary File 3 to which you refer in your text on page 22. Please amend the file type to 'Supporting Information'. If the Supplementary file is no longer to be included as part of the submission please remove all reference to it within the text.

AUTHORS: Thank you for the suggestion. We have revised based on the PLoS One formatting guidelines https://journals.plos.org/plosone/s/file?id=wjVg/PLOSOne_formatting_sample_main_body.pdf

Additional Editor Comments:

Please briefly elaborate your argument about the possible link between your case as risk & the mentioned diseases. If possible, following a theoretical framework may make sense of your findings.

AUTHORS: Thank you for the suggestion. Our study is an exploration of experience of living in Star Homes and we used a descriptive qualitative method to explore their everyday interactions with the parts of the house, spaces, and changes in social dynamics brought by the interventions. In the sixth paragraph in introduction section, we have outlined how our primary study is to evaluate the impact of through randomized controlled trial comparing the incidence of the three diseases (malaria, ARIs, and diarrheal diseases) in children under 13 years living in Star Homes compared with traditional wattle-daub houses. 

Reviewers' comments:

Reviewer #1: There are issues with how this article has been written, organised and structured, overall. I suggest going through PLOS ONE guidelines for structuring the article to begin with.

- Introduction section is too loaded with the information about why novel houses are needed in the community. This section will be more balanced with maximum 2 short paragraphs on their relevance in preventing mentioned diseases, and the rest on how in other similar contexts communities have responded to living in tailored houses.

AUTHORS: Thank you for the suggestion. We agree with your suggestion. We broadly followed PLoS One’s instruction for authors’ guideline for manuscript organization.

We agree that ‘introduction’ section looks loaded with multiple paragraphs. As you are aware that the study intervention has multi-pronged impacts (multi-disciplinary dimensions, impacts on health conditions, living space, social dynamics and feasibility), we wanted to include an overarching background and rationale in the ‘introduction’ section to consolidate our focus on the main intervention that is on infectious diseases, and forge how our current manuscript aligns with the main aim by exploring the experience of living in the star homes. 

-The sub sections in the Methods section needs to be titled respectively as Study Setting, Sampling, Data Collection, Data Management and Analysis and Ethical Considerations. The sub-section, Study Design seems an unnecessary addition. The description of who did what can easily fit into the suggested subtopics. Also, mentioning names of architect firms with their links in the bracket looked amateurish to me.

AUTHORS: Thank you for the suggestion. We agree with your suggestion. We have revised the order of respective sections and the content. As you may be aware that there is heterogeneity in organizations of titles within the published articles, we have incorporated your suggestion that best suits our manuscript. 

With the ‘Study design’ as a sub-topic within the materials and methods, we request reviewer to consider our following justification. 

We would like to retain a broader overview of our study in terms of how it is organized. Although the design and overview could be inserted into other sub-sections, having it clearly labelled and described it may serve the reader a quick glance of the study’s overarching methodology. 

Regarding mentioning the name of architect firm with their links; although broadly applicable, given their critical role in designing the Star Homes, and following the tradition how architectural firms are required to be explicitly reported, we included the name and the link within the manuscript. In the prior publication as well, we have reported in a similar format [1]. 

- I suggest not to provide an overview of Results and directly get to the point. The Overview anyway presents a description of collected data, which should fit into the Data Collection section.

AUTHORS: Thank you for the suggestion. We have revised and added the relevant content under the ‘Data collection’ section. 

- Illustration of case studies looked abrupt to me, as the article did not mention anywhere that individual cases were studied, and even if it had, there needed to be clear mentioning of sampling procedure for the selection of those cases.

AUTHORS: Thank you for the suggestion. We have revised the section ‘sampling’ to add how we have selected these four Star Homes cases in this study. The revised section has additional statements to clarify this and reads as follows: 

The interviews were followed by observations and informal interactions with a total of four Star Homes to document the diverse spectrum of experience of living in the Star Homes. These four Star Homes were purposively selected based on the distinctive issues and problems faced by the residents.

- In some instances, the result section jumps to conclusion without formulating proper argument to back it up. Result section should not present any discussion or conclusion. For e.g, p17: “Some respondents reported neighbors made comments that the Star Home residents were living on aid and made negative comments such as the houses were only for Christians. Such comments were manifestations of jealousies.” To assert someone’s opinion as jealousy is not a very academic way of sharing findings. Authors could have said (if applicable), “Respondents said they sensed jealousy in their neighbor's comments", and back it up with a quote. further, I should avoid discussing Jealousy as one of the topics in the Discussion. Please rephrase it as something similar to “Unfavorable comments from the neighbors”. The neighborhood not getting Star Houses could have (reasonably) triggered certain reactions but labeling them as jealous people makes the findings look biased.

AUTHORS: Thank you for the suggestion. We agree with you, that we asserted them as manifestation of jealousy, although such manifestations are often the envy rather than jealousy, they have been widespread and common in such interventions including randomized clinical trials [2-6]. We have revised the statement and have removed our statement (‘such comments were manifestations of jealousies) that looked assertive. 

Re-adding the quotes, comments and feedback implying envy were often found during our semi-structured interviews (the first conversation under ‘response of the wider community’), observational field notes, including case studies. For instance, case study-I and IV have such comments and expressions that alluded towards the envy, so we attempted to avoid the redundancy by adding additional quotes. 

Regarding jealousy as a major theme in ‘discussion’ section, we are very thankful to you for raising this, as it allowed us to reflect on it more and explore more literature around it. We have revised the section as ‘envy’ and please consider our justification bellow: 

First, envy is one of the commonest theme in the literature on how selective participation in research and interventions creates division. Second, envy in our research intervention is an inductively derived theme thus we wanted to retain it. Previous literature have identified such themes (both envy and jealousy) and have presented in their findings as well [2-6]. Third, ‘envy’ in our research intervention has been identified during our iterative data analysis and has been a prominent finding during our longitudinal data collection phases. Fourth, we are aware of the connotations borne by the word ‘envy’ (and jealousy)[7], and our position in this study is not to label the non-participants negatively as envious but authentically reporting it as it emerged from our data. As you may be aware that envy has a positive connotation too; that is desiring to achieve something or desiring to receive a tempting incentive. In addition, ‘envy’ is a natural pan-human attribute/manifestation[7] when select of community members receive an intervention (Star Home is a major incentive or receiving health services attached to participating in a clinical trial) which is irresistible. 

Reviewer #2: • Please explain how "Star Home" protects its residents from malaria, ARIs, and diarrheal infections.

AUTHORS: Thank you for raising this question. This is also our main research question related to our intervention: Star Homes, and we do not yet have data to report. We have revised a statement in the manuscript within the sub-section ‘Internal and external spaces’ within Discussion that reads as follows: 

Such research can potentially contribute to explaining the impact (or lack thereof) of housing interventions on health outcomes such as malaria, respiratory tract infections and diarrheal diseases and is currently being evaluated by the clinical study [8, 9].

• Mention the criteria for recruiting respondents.

AUTHORS: Thank you for the suggestion. We have revised how we recruited the respondents in this study under the revised heading ‘sampling’ within ‘materials and methods’ section that reads as follows: 

Residents of Star Homes and traditional homes were recruited for the current qualitative study because of issues they had reported to the researc

---

## [Decision Letter · Decision Letter 1]

7 Jul 2024

PONE-D-24-05994R1Community responses to a novel house design: a qualitative study of “Star homes” in Mtwara, southeastern TanzaniaPLOS ONE

Dear Dr. Adhikari,

Thank you for submitting your manuscript to PLOS ONE. After careful consideration, we feel that it has merit but does not fully meet PLOS ONE’s publication criteria as it currently stands. Therefore, we invite you to submit a revised version of the manuscript that addresses the points raised during the review process.

We look forward to receiving your revised manuscript.

Kind regards,

Yitagesu Habtu Aweke, Ph.D

Academic Editor

PLOS ONE

Journal Requirements:

**Additional Editor Comments:**

Could you please address “steps to maintaining rigors of the study” so that scientific rigors in our journal may be achieved? I have some concerns specifically on methodological rigors such as:  **  **

It seems that adequate study participants were participated in both IDIs and FGDs, but supporting quotes in each theme are very few, I could say, not more than 1 for majority of your themes. In this instance, the most important rigor criteria of “think descriptions” seems to be affected.  Could you please attach at least sample of participants’ quotations supporting your key findings in each them in the form of at least a “supplementary files”?Could you show how you maintain methodological rigors, and where you put findings in each theme?

Reviewer's Responses to Questions

**Comments to the Author**

1. If the authors have adequately addressed your comments raised in a previous round of review and you feel that this manuscript is now acceptable for publication, you may indicate that here to bypass the “Comments to the Author” section, enter your conflict of interest statement in the “Confidential to Editor” section, and submit your "Accept" recommendation.

Reviewer #1: All comments have been addressed

2. Is the manuscript technically sound, and do the data support the conclusions?

Reviewer #1: Yes

3. Has the statistical analysis been performed appropriately and rigorously? 

Reviewer #1: Yes

4. Have the authors made all data underlying the findings in their manuscript fully available?

Reviewer #1: No

5. Is the manuscript presented in an intelligible fashion and written in standard English?

Reviewer #1: Yes

6. Review Comments to the Author

Reviewer #1: (No Response)

7. PLOS authors have the option to publish the peer review history of their article (what does this mean?). If published, this will include your full peer review and any attached files.

Reviewer #1: No

---

## [Author Response · Author response to Decision Letter 1]

13 Jul 2024

10th July 2024

Dear Prof. Yitagesu Habtu Aweke, 

Thank you very much for your suggestion and opportunity to revise our work. We have utilized your suggestions to improve our manuscript. Please see below our responses to your specific suggestions with corresponding changes in the manuscript. 

We look forward to your kind consideration. 

On behalf of all the authors,

Bipin Adhikari

Additional Editor Comments:

Could you please address “steps to maintaining rigors of the study” so that scientific rigors in our journal may be achieved? I have some concerns specifically on methodological rigors such as: 

• It seems that adequate study participants were participated in both IDIs and FGDs, but supporting quotes in each theme are very few, I could say, not more than 1 for majority of your themes. In this instance, the most important rigor criteria of “think descriptions” seems to be affected. Could you please attach at least sample of participants’ quotations supporting your key findings in each them in the form of at least a “supplementary files”?

AUTHORS: Thank you for your evaluation and useful feedback. In the manuscript, we tried to present as succinctly as possible to ensure that we also accommodate four case studies that offer thick description on living experience (social and cultural context) in Star Homes. Following your suggestion, we have also added a sample of participants’ conversations containing quotations (from our latest data collection phase) supporting the key findings and are presented in a supplementary file 3. 

• Could you show how you maintain methodological rigors, and where you put findings in each theme?

AUTHORS: Thank you for the question. Aligned with your question/suggestion above, we have presented codebook, sunburst visualization of coding landscape and supporting quotes for themes included in the manuscript in the supplementary file 3.

---

## [Editor Report · Decision Letter 2]

17 Jul 2024

PONE-D-24-05994R2Community responses to a novel house design: a qualitative study of “Star homes” in Mtwara, southeastern TanzaniaPLOS ONE

Dear Dr.  Adhikari,

Thank you for submitting your manuscript to PLOS ONE. After careful consideration, we feel that it has merit but does not fully meet PLOS ONE’s publication criteria as it currently stands. Therefore, we invite you to submit a revised version of the manuscript that addresses the points raised during the review process.Indicate which changes you require for acceptance versus which changes you recommendAddress any conflicts between the reviews so that it's clear which advice the authors should followProvide specific feedback from your evaluation of the manuscript

Please include the following items when submitting your revised manuscript:A rebuttal letter that responds to each point raised by the academic editor and reviewer(s). You should upload this letter as a separate file labeled 'Response to Reviewers'.A marked-up copy of your manuscript that highlights changes made to the original version. You should upload this as a separate file labeled 'Revised Manuscript with Track Changes'.An unmarked version of your revised paper without tracked changes. You should upload this as a separate file labeled 'Manuscript'.If applicable, we recommend that you deposit your laboratory protocols in protocols.io to enhance the reproducibility of your results. Protocols.io assigns your protocol its own identifier (DOI) so that it can be cited independently in the future. For instructions see: https://journals.plos.org/plosone/s/submission-guidelines#loc-laboratory-protocols. Additionally, PLOS ONE offers an option for publishing peer-reviewed Lab Protocol articles, which describe protocols hosted on protocols.io. Read more information on sharing protocols at https://plos.org/protocols?utm_medium=editorial-email&utm_source=authorletters&utm_campaign=protocols.

We look forward to receiving your revised manuscript.

Kind regards,

Yitagesu Habtu Aweke, Ph.D

Academic Editor

PLOS ONE

Journal Requirements:

Additional Editor Comments:Could you please summarize  supplementary file 3   in the form  of  single table containing column labels as  ' "Codes", "Themes", "sample quotes" ' ? Not the full  transcriptions and full codebook" 

---

## [Author Response · Author response to Decision Letter 2]

25 Jul 2024

18th July 2024

Dear Prof. Yitagesu Habtu Aweke, 

Thank you very much for your specific and clear suggestion and opportunity to revise our work. We have followed your suggestions to revise the supplementary Table. Please see below our responses to your specific suggestions with corresponding changes in the manuscript. 

We look forward to your kind consideration. 

On behalf of all the authors,

Bipin Adhikari

Additional Editor Comments:

• Could you please summarize supplementary file 3 in the form of single table containing column labels as ' "Codes", "Themes", "sample quotes" ' ? Not the full transcriptions and full codebook" 

AUTHORS: Thank you for your specific suggestion. We have prepared the supplementary Table 3 based on your specific suggestions.

---

## [Editor Report · Decision Letter 3]

15 Aug 2024

Community responses to a novel house design: a qualitative study of “Star homes” in Mtwara, southeastern Tanzania

PONE-D-24-05994R3

Dear Dr. Bipin Adhikari,

We’re pleased to inform you that your manuscript has been judged scientifically suitable for publication and will be formally accepted for publication once it meets all outstanding technical requirements.

Kind regards,

Yitagesu Habtu Aweke, Ph.D

Academic Editor

PLOS ONE

---

## [Editor Report · Acceptance letter]

19 Aug 2024

PONE-D-24-05994R3 

PLOS ONE

Dear Dr. Adhikari, 

I'm pleased to inform you that your manuscript has been deemed suitable for publication in PLOS ONE. Congratulations! Your manuscript is now being handed over to our production team.

Kind regards, 

on behalf of

PhD Candidate Yitagesu Habtu Aweke 

Academic Editor

PLOS ONE